# APOLLO: ZERO-SHOT MULTIMODAL REASONING WITH MULTIPLE EXPERTS

## ABSTRACT

We propose a modular framework that leverages the expertise of different foundation models over different modalities and domains in order to perform a single, complex, multi-modal task, without relying on prompt engineering or otherwise tailor-made multi-modal training. Our approach enables decentralized command execution and allows each model to both contribute and benefit from the expertise of the other models. Our method can be extended to a variety of foundation models (including audio and vision), above and beyond only language models, as it does not depend on prompts. We demonstrate our approach on two tasks. On the well-known task of stylized image captioning, our experiments show that our approach outperforms semi-supervised state-of-the-art models, while being zero-shot and avoiding costly training, data collection, and prompt engineering. We further demonstrate this method on a novel task, audio-aware image captioning, in which an image and audio are given and the task is to generate text that describes the image within the context of the provided audio.

## 1 INTRODUCTION

Humans perceive the world through different types of data (e.g., images and sounds) that they get from their senses. Similarly, to understand the world, artificial intelligence research also tries to solve problems that use multimodal data (Antol et al., 2015; Paz-Argaman et al., 2020; Ji et al., 2022; Rassin et al., 2023). Solving multimodal tasks requires interpreting and reasoning over heterogeneous data, which poses several challenges, such as the training process (Wang et al., 2020).

Large pre-trained *foundation* models demonstrate distinct expertise and encompass comprehensive knowledge within specific domains and modalities they are trained on. For example, BERT (Devlin et al., 2018) and GPT3 (Brown et al., 2020)are proficient in processing language, while CLIP (Radford et al., 2021) excels in grounding text to visual content. However, the large and increasing variety of multimodal tasks (e.g., vision and language navigation (Ku et al., 2020), and video question-answering (Lei et al., 2018)), do not have *foundation* models. Previous efforts to tackle complex multimodal tasks are either (1) fully-supervised, require expensive paired input and output task-specific data (Chen et al., 2019; Li et al., 2022);(2) semi-supervised – task-specific uncoupled data for each modality or domain (Nukrai et al., 2022; Guo et al., 2019; Zhao et al., 2020; Gan et al., 2017; Su et al., 2022); (3) few-shot – a few coupled task-specific examples; and (4) Zero-shot (ZS) – no task-specific data. The approaches for ZS contain a sequence-to-sequence unified approach that is trained on multiple tasks (Lu et al., 2022; Zhu et al., 2022; Gupta et al., 2022). However, as the list of tasks is fixed, so any new task requires changes to the model and additional training.

*Socratic models*, an approach for few-shot and ZS learning, composes pre-trained models by directly using language as the intermediate representation by which the modules exchange information with each other(Zeng et al., 2022). Thus, this approach heavily relies on a large language model (LLM) and requires prompt engineering which does not have a proper methodology. Relying on LLMs might be sub-optimal, particularly for multimodal tasks that do not involve language, e.g., music and vision tasks (Qiu & Kataoka, 2018; Aleixo et al., 2021).

In this paper, we propose a different approach to multimodal tasks that leverages the expertise of foundation models and shares knowledge through a common latent space without relying on language as a mediator. The importance of knowledge sharing between experts can be illustrated by the Apollo program, which required the collaboration of experts from diverse fields, such as physics, chemistry,

and biology, to achieve the common goal of landing a man on the moon. By sharing their knowledge, these experts were able to overcome the challenges and undertake a task never done before. Our premise that complex tasks, like the Apollo, require multiple experts, inspired our approach which relies on synergy and knowledge sharing between pre-trained transformer components through gradient updating of a combined loss at inference time. This allows our model to perform new tasks in a zero-shot setup without any further training or tuning steps. Unlike Socratic models, the proposed framework, which we named APOLLO, is not limited to language models. It can be applied to a variety of transformer models of different modalities, such as audio and vision, moving beyond LLMs and not depending on prompts. Furthermore, APOLLO enables decentralized command execution, allowing each model to contribute and benefit from the expertise of others.

We demonstrate our approach on two tasks. On the well-known task of stylized image captioning (Zhao et al., 2020; Guo et al., 2019; Nukrai et al., 2022; Mathews et al., 2016; Gan et al., 2017), our ZS Apollo method gained an absolute improvement of up to 58% in style accuracy and up to 2.3% in relevance text to the image, compared to the state-of-the-art semi-supervised models on the SentiCap (Mathews et al., 2016) and FlickrStyle10K (Gan et al., 2017) benchmarks. We further demonstrate this method on a novel task, audio-aware image captioning, in which an image and audio are given and the task is to generate text that describes the image within the context of the provided audio.

## 1.1 THE APOLLO METHODS

The cutting-edge models across diverse modality domains primarily rely on transformer-based architectures (Vaswani et al., 2017). Our objective is to leverage the expertise of multiple pre-trained transformer models to generate output through shared impact between the models. A Transformer model consists of two primary components: an encoder and a decoder. Each component comprises $L$ layers of encoders and decoders, and within these layers, multiple attention heads are present, each with query ($Q$), key ($K$), and value ($V$) functions. The attention mechanism enables the model to selectively focus on different parts of the input data. This focus is determined by the interactions between $Q$ and $K$, which produce attention scores and influence the distribution of $V$. Function $Q$ operates on the input token embedding, while $K$ and $V$ generate subsequent output tokens by considering past tokens. This implies that both the $K$ and the $V$ can influence the final prediction output, given $Q$. To exercise control over the model's output, we seek to influence the 'context cache', which contains both the key ($K$) and the value ($V$), thus guiding the model's predictions towards a desired direction. We consider a probability vector for the output of a transformer model $T_j$: $P_{T_j}(\{x_i\}_{i=1}^n | \{m_i\}_{i=1}^M; T_j(\cdot|C_{T_j}^l))$, where $P_{T_j}$ represents the probability of candidates $\{x_i\}_{i=1}^n$ conditioning on modalities $\{m_i\}_{i=1}^M$. The probability is parameterized by an expert transformer $T_j$ for which we select a subset of $K$ and $V$ from certain layers $l$ to define a context, $C_{T_j}^l$.

**Two Experts** We generalize the loss function used by Tewel et al. (2021) to any two transformer models where transformer $T_1$ shares knowledge with $T_2$. We get the following loss:

$$\mathcal{L} \triangleq CE(P_{T_2}^{(t)}, P_{T_1}) + \lambda \cdot CE(P_{T_2}^{(t)}, P_{T_2}^{(0)}) \tag{1}$$

In order to guide the model's prediction, we minimize the loss in equation 1 over the context $C_{T_2}$, which implements the following concept: The first term in equation 1 pulls the preference tokens of transformer $T_2$ towards the target token preferences of $T_1$ through $t$ gradient steps, potentially overriding the original knowledge of transformer $T_2$. To preserve the transformer's original knowledge, an additive regularization term constrains the transformer's deviation from its initial preference, $P_{T_2}^{(0)}$. $\lambda$ is a hyper-parameter that balances the two loss terms. The guidance method implemented by equation 1 is denoted as *Experts-Summation*.

**Multiple Experts** We consider a framework that contains M≥2 expert-transformers $\{T_j\}_{j=1}^M$. The *Experts-Summation* can be extended to the multi-expert case by simply summing multiple weighted terms in the loss function: $\mathcal{L} = CE(P_{T_M}^{(t)}, P_{T_M}^{(0)}) + \Sigma_{j=1}^{M-1} \lambda_j \cdot CE\left(P_{T_M}^{(t)}, P_{T_j}\right)$. This extension comes at the cost of tuning multiple hyper-parameters, making it challenging to find the balance between all experts' loss components. Therefore, we propose a new guidance loss inspired by the

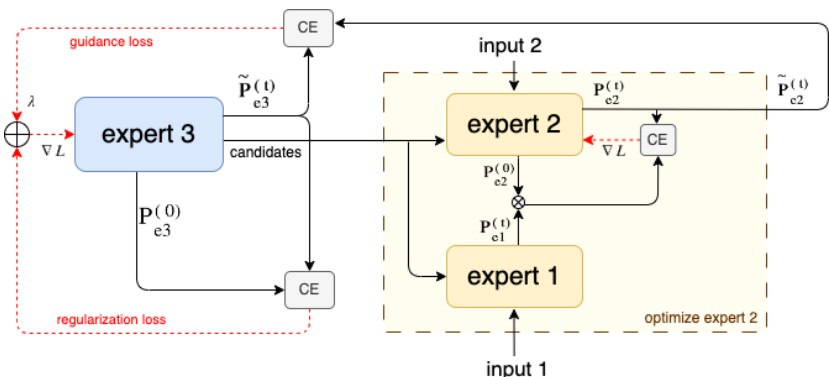

Figure 1: An overview of *Decentralization of Guidance Efforts* approach.

attention concept, which offers a safer alternative – *Experts-Product*:

$$\mathcal{L} = CE(P_{T_M}^{(t)}, P_{T_M}^{(0)}) + \lambda \cdot CE\left(P_{T_M}^{(t)}, \Pi_{j=1}^{M-1} P_{T_j}\right) \tag{2}$$

The target probability in the second term of equation 2 is the element-wise multiplication of all experts' probabilities, denoted as $\{P_{T_j}\}_{j=1}^{M-1}$. This operation merges the experts' preferences and directs the transformer $T_M$ toward a common region, while maintaining proximity to the initial suggestion boundaries, as guided by the first loss term. It does not add hyper-parameters comparing to *Experts-Summation* and yet it effectively enforces $P_{T_M}^{(t)}$ to agree with the experts common support.

**Decentralization of Guidance Efforts**    In the case of M expert-transformers, the straightforward way to apply all the experts' preferences to $P_T$ is by optimizing a flat objective function, as in *Expert-Product*. One challenge in accommodating all preferences simultaneously is the lack of effective communication among the guiding experts themselves. Alternatively, we propose a hierarchical optimization process, in which one domain expert guides another, and the latter guides the top-level expert model. This allows experts to share their knowledge not only with the top-level expert model but also with each other. In this process, a mediator expert is responsible for producing the final recommendation for the top-level expert model. This expert considers the perspective of the other experts and adapts to minimize potential conflicts in their guidelines. To better understand this approach, we demonstrate it on a case of $M = 3$ expert transformers as presented in Figure 1. In this example, expert 1 ($e_1$) and 2 ($e_2$) are domain-experts who guide a top-level expert - ($e_3$) which plays a central role in the system. $P_{e_1}, P_{e_2}, P_{e_3}$ denote the probabilities for the candidates $\{x_i\}_{i=1}^n$ over experts 1,2,3 respectively. The objective of aligning Expert 3 ($P_{e_3}$) with both Expert 1 ($P_{e_1}$) and Expert 2 ($P_{e_2}$) is achieved by solving the hierarchical optimization problems defined by the following equations:

$$\tilde{C}_2^{(t)} = \underset{C_2}{argmin}\{CE(P_{e_2}^{(t)}, P_{e_2}^{(0)} \cdot P_{e_1}^{(t)})\} \tag{3}$$

$$\tilde{C}_3^{(t)} = \underset{C_3}{argmin}\{CE(P_{e_3}^{(t)}, P_{e_3}^{(0)}) + \lambda \cdot CE(P_{e_3}^{(t)}, \tilde{P}_{e_2}^{(t)})\} \tag{4}$$

First, we optimize the probability $P_{e_2}$ over context cache $C_2$ (equation 3). Second, we optimize the model probability $P_{e_3}$ at the top-hierarchy by adjusting $C_3$ (equation 4). This approach decentralizes the guidance efforts among multiple models, enhancing the interaction between the experts.

## 2   STYLIZED IMAGE CAPTION GENERATION

**Goal**    Our objective in this task is to generate captions that accurately describe the input image while incorporating the desired style. We aim to achieve this without training any model. Instead, our approach focuses on leveraging the expertise of diverse models and utilizing their capabilities to generate captions with the desired style.

## 2.1 METHOD

**APOLLO-CAP**  In order to generate captions for images with a specific style, we use multiple experts. We use the LLM GPT-2 (Radford et al., 2019) to iteratively predict tokens. We use GPT-2 instead of its advanced versions, e.g., GPT-3, because GPT-2 is open-source, allowing us to modify its internal representations, such as its keys (Q) and values (V). We use an image-text alignment model – CLIP (Radford et al., 2021) to evaluate the relevance of each candidate token to the given image. Each candidate token is appended to the current partial sentence ($X_{t,i} = x_{i+1}^{(t)}, x_i, ..., x_0$), and combined with the image as input to CLIP. The cosine similarity $S^{CLIP}$ between each candidate and the image is computed in the embedding space, and probabilities are generated by applying softmax with a smoothing temperature parameter $\tau$.

$$\boldsymbol{S}_{x_{i+1}}^{CLIP} = S^{CLIP}(\boldsymbol{X}_{t,i}, I)$$
$$\boldsymbol{p}_{x_{i+1}}^{CLIP} \triangleq softmax(\boldsymbol{S}_{x_{i+1}}^{CLIP}; \tau) \tag{5}$$

We consider the first layer output K and V as CLIP's *context* for guidance purposes. Our last expert is a Style-Text Alignment We employ a style classification model and score each candidate based on its alignment with the desired style. We generate probabilities for all candidates by applying softmax with a smoothing temperature parameter. We use roBERTa (Liu et al., 2019) for sentiment realization and DeepMoji (Felbo et al., 2017) for applying romantic and humorous style.

$$\boldsymbol{S}_{x_{i+1}}^{STYLE} = S^{STYLE}(\boldsymbol{X}_{t,i}|STYLE)$$
$$\boldsymbol{p}_{x_{i+1}}^{STYLE} \triangleq softmax(\boldsymbol{S}_{x_{i+1}}^{STYLE}; \tau) \tag{6}$$

---

**Algorithm 1:** Optimizing GPT-2 towards an image and style

1 **for** *i=0...* **do**
2     **for** *t=0...T-1* **do**
3        $\boldsymbol{p}_{x_{i+1}}^{(t)} \leftarrow GPT(\boldsymbol{x}_i, C_i^{(t)})$
4        $\boldsymbol{X}_{t,i} \leftarrow \boldsymbol{x}_{i+1}^{(t)}, x_i, ..., x_0$
5        $\{\boldsymbol{p}_{x_{i+1}}^{<expert>}\}$
         $\leftarrow$calc_probability$(\boldsymbol{X}_{t,i}, I, STYLE)$
6        $\mathcal{L}$
         $\leftarrow$calc_loss$(\boldsymbol{p}_{x_{i+1}}^{(t)}, \{\boldsymbol{p}_{x_{i+1}}^{<expert>}\}, \boldsymbol{p}_{x_{i+1}}^{(0)})$
7        $\mathrm{C}_i^{(t+1)} \leftarrow C_i^{(t)} + \alpha \frac{\nabla_{C_i}\mathcal{L}}{\|\nabla_{C_i}\mathcal{L}\|^2}$
8     **end**
9     $x_{i+1} \leftarrow \underset{\boldsymbol{x}}{argmax} GPT(x_i, C_i^{(T)})$
10     **if** $x_{i+1} = EndToken$ **then**
11        break
12     **end**
13 **end**

**Algorithm 2:** Optimizing CLIP image embedding towards the desired style

1 Initialize $C^{(0)}$ to CLIP's default context
2 **for** *j=0...J-1* **do**
3     $\boldsymbol{P}_{CLIP}^{(j)} = CLIP(\boldsymbol{x}, I|C^{(j)})$
4     $\mathcal{L} \leftarrow CE\left(\boldsymbol{P}_{CLIP}^{(j)}, \boldsymbol{P}_{target}\right)$
5     $\mathrm{C}^{(j+1)} \leftarrow C^{(j)} + \alpha \frac{\nabla_C\mathcal{L}}{\|\nabla_C\mathcal{L}\|^2}$
6 **end**
7 **return** $CLIP(\boldsymbol{x}, I|C^{(J)})$

---

## 2.2 GRADIENT UPDATES FOR MODEL GUIDING

By combining the image-oriented and style-oriented probabilities, we can manipulate GPT-2 through its context vector to generate an image caption with the desired style.

Let $I$ be the input image, $x_{i+1}$ the next candidate token, $C_i$ the GPT-2's context vector, and $p_{x_{i+1}} = GPT(x_i, C_i)$ the probability predicted by GPT-2 for $x_{i+1}$. The goal is to iteratively optimize the context $C_i$ in order to improve the description of the image with the desired style. The optimization steps are outlined in Algorithm 1. For each generated token, a total of $T$ optimization steps are performed as follows: An alternative probabilities of the next token are calculated according to a set

of experts (row 5). Then, a loss function is computed incorporating the experts prediction (row 6). As suggested by ZeroCap (Tewel et al., 2021), a regularization term is added to keep the optimized probability close to the original probability generated by GPT-2 in the initial step. Minimizing this loss over the context vector results in an image-style-aware probability. The context vector is updated by applying a single gradient step (row 7). This optimization loop is repeated for each generated token until the captioning process is complete. The outer loop is executed with 5 beams, and the inner loop is applied to the top K=512 tokens.

Next, we provide a detailed implementation for each guidance approach described in Section1.1.

**APOLLO-CAP: Sum of Experts** After the generative transformer calculates its probability for the next token, each expert calculates its alternative probability. To align image and text, we calculate the CLIP probability $\boldsymbol{p}_{x_{i+1}}^{CLIP}$ for the top 512 candidates (see equation 5) to determine the best probability vector for image-text correspondence. In addition, style-aware probability $\boldsymbol{p}_{x_{i+1}}^{STYLE}$ is computed based on the style model's scores to encourage a certain style (see equation 6). The guidance loss $\mathcal{L}$ is computed as a weighted sum of the cross-entropy between the augmented probabilities and the baseline GPT-2 probability:

$$\mathcal{L} = \lambda_{LM}CE\left(\boldsymbol{p}_{x_{i+1}}^{(t)}, \boldsymbol{p}_{x_{i+1}}^{(0)}\right) + \lambda_{CL}CE\left(\boldsymbol{p}_{x_{i+1}}^{(t)}, \boldsymbol{p}_{x_{i+1}}^{CLIP}\right) + \lambda_{SL}CE\left(\boldsymbol{p}_{x_{i+1}}^{(t)}, \boldsymbol{p}_{x_{i+1}}^{STYLE}\right) \quad (7)$$

**APOLLO-CAP: Product of Experts** Similarly to sum of experts, CLIP probability $\boldsymbol{p}_{x_{i+1}}^{CLIP}$ and style probability $\boldsymbol{p}_{x_{i+1}}^{STYLE}$ are computed according to equation 5 and equation 6 respectively. The guided loss $\mathcal{L}$ is composed of two terms: (1) the cross entropy between the product of CLIP and STYLE probabilities with the current GPT suggestion, and (2) a regularization term:

$$\mathcal{L} = \lambda_{LM}\underbrace{CE\left(\boldsymbol{p}_{x_{i+1}}^{(t)}, \boldsymbol{p}_{x_{i+1}}^{(0)}\right)}_{regularization} + \lambda_{CL}\underbrace{CE\left(\boldsymbol{p}_{x_{i+1}}^{(t)}, \boldsymbol{p}_{x_{i+1}}^{CLIP} \cdot \boldsymbol{p}_{x_{i+1}}^{STYLE}\right)}_{experts} \quad (8)$$

**APOLLO-CAP: Decentralization** We suggest optimizing CLIP's image embedding such that the resulting text-image matching will be more style-oriented. Since CLIP is a transformer encoder, we apply the decentralization concept described in Section 1.1 as follows: Let $\boldsymbol{x}$ be candidate captions for image $I$. We denote CLIP's first layer $K, V$ outputs by $C$ as context vector for optimization. Let $\boldsymbol{P}_{STYLE}$ be the probability vector produced by the style expert model given $\boldsymbol{x}$, and $\boldsymbol{P}_{CLIP}^{(0)} = CLIP(\boldsymbol{x}, I|C^{(0)})$ be CLIP's initial probability prediction for $\boldsymbol{x}$ given $I$ conditioning on the initial context $C^{(0)}$. We compute the target probability as the product of the style expert probability and CLIP's initial probability: $\boldsymbol{P}_{target} = \boldsymbol{P}_{CLIP}^{(0)} \cdot \boldsymbol{P}_{STYLE}$. We apply $J$ gradient steps to optimize CLIP's image embedding. As a result, the optimized CLIP produces higher probabilities for captions that fit the image content from the specific style perspective. This approach is presented in Algorithm 2. We denote the output probability as $\boldsymbol{p}_{x_{i+1}}^{CLIP-STYLE}$, and then incorporate it into the loss function presented in equation 8, resulting in the guidance loss $\mathcal{L}$:

$$\mathcal{L} = \lambda_{LM}CE\left(\boldsymbol{p}_{x_{i+1}}^{(t)}, \boldsymbol{p}_{x_{i+1}}^{(0)}\right) + \lambda_{CL}CE\left(\boldsymbol{p}_{x_{i+1}}^{(t)}, \boldsymbol{p}_{x_{i+1}}^{CLIP-STYLE} \cdot \boldsymbol{p}_{x_{i+1}}^{STYLE}\right) \quad (9)$$

### 2.3 EXPERIMENTAL SETUP

**Data** We evaluate our approach on the two benchmarks, SentiCap (Mathews et al., 2016) for positive and negative styling and FlickrStyle10K (Gan et al., 2017) for humor and romantic.

**Evaluation Metrics** To evaluate the results, we examined the following attributes of the captions: (1) *fluency*, i.e., the coherency and naturalness of the generated text; (2) *Text-Image correspondence (TIC)*, and (3) *style accuracy*. We evaluate *fluency* using the perplexity function of GPT-2, which measures the model's ability to predict the next word in a sequence. Lower perplexity values indicate better fluency of the generated captions. The perplexity scores were clipped to the maximal value of 1500 and then normalized by $1 - \dfrac{perplexity}{1500}$, formalizing a fluency score (the higher the better).

| Model | positive | | | | negative | | | |
|---|---|---|---|---|---|---|---|---|
| | TIC | style accuracy | fluency | Vocab | TIC | style accuracy | fluency | Vocab |
| CapDec | 0.294 | 0.53 | **0.97** | 717 | 0.292 | 0.3 | **0.97** | 706 |
| ZeroCap+PM | 0.327 | 0.79 | 0.93 | 2715 | 0.31 | 0.79 | 0.94 | 2937 |
| ZeroCap+IM | **0.328** | 0.24 | 0.84 | **2917** | **0.33** | 0.13 | 0.83 | 2884 |
| ZeroCap+IPM | 0.327 | 0.86 | 0.93 | 2736 | 0.312 | 0.8 | 0.94 | **3025** |
| APOLLO-CAP | 0.268 | 0.91 | 0.9 | 1978 | 0.267 | 0.76 | 0.86 | 2417 |
| APOLLO-CAP-P | 0.283 | **0.97** | 0.84 | 1658 | 0.291 | **0.88** | 0.85 | 2302 |
| APOLLO-CAP-PD | 0.317 | 0.94 | 0.8 | 2200 | 0.296 | 0.81 | 0.85 | 2544 |

| Model | humorous | | | | romantic | | | |
|---|---|---|---|---|---|---|---|---|
| | TIC | style accuracy | fluency | Vocab | TIC | style accuracy | fluency | Vocab |
| CapDec | 0.285 | 0.05 | **0.98** | 885 | 0.285 | 0.12 | **0.98** | 822 |
| ZeroCap+PM | 0.325 | 0.06 | 0.88 | 2875 | 0.321 | 0.09 | 0.87 | **2983** |
| ZeroCap+IM | **0.326** | 0.05 | 0.81 | 2818 | **0.325** | 0.07 | 0.8 | 2855 |
| ZeroCap+PIM | 0.325 | 0.07 | 0.93 | 2531 | 0.317 | 0.14 | 0.92 | 2728 |
| APOLLO-CAP | 0.269 | 0.06 | 0.91 | **3001** | 0.268 | 0.13 | 0.84 | 2712 |
| APOLLO-CAP-P | 0.286 | **0.23** | 0.90 | 2444 | 0.262 | 0.32 | 0.88 | 2621 |
| APOLLO-CAP-PD | 0.298 | 0.2 | 0.85 | 2774 | 0.28 | **0.37** | 0.81 | 2527 |

*TIC- text-image correspondence

Table 1: Averaged Scores for CapDec, ZeroCap Manipulations and Apollo-Cap Approaches
ZeroCap best fits the image content, but fails to generate style with only input manipulations. CapDec as a semi supervised model for image captioning shows fluent language but also weaker style realization.
APOLLO-CAP-PD outperforms the other approaches in the total image-text-style matching trade-off.

In order to quantify the alignment between an image and its caption (*TIC*), we used CLIPScore (Hessel et al., 2022) - the cosine similarity between the CLIP embedding of the image and the caption. We measured *Style Accuracy* using large pre-trained models – roBERTa (Hartmann et al., 2022) and DeepMoji (Felbo et al., 2017). roBERTa is a sentiment classification model that generates probability for either positive or negative. In order to evaluate the emotional styles of Flickrstyle10k – humorous and romantic, we employed DeepMoji model. Given a text input, DeepMoji generates a 64-dimensional probability vector for various emotions which are aggregated to represent humorous and romantic styles (see Appendix A.3).

**Models** We demonstrated the three zero-shot methods described in Sections 2.2 by plugging-in the loss functions in equation 7, equation 8, equation 9 into ZeroCap as drop-and-replace of its original loss. Specifically, we employed the techniques *Experts-Summation* which will be referred to as APOLLO-CAP, *Expert-Product* (APOLLO-CAP-P) and combination of *Decentralization of Guidance Efforts* with *Expert-Product* (APOLLO-CAP-PD).

**Baselines** We conducted a comparative analysis of our method with the current state-of-the-art technique for generating stylized image captions, namely CapDec (Nukrai et al., 2022). CapDec, a semi-supervised method, relies on training a decoder using stylized text to generate stylized captions. It achieves this by leveraging the shared embedding space of text and images in CLIP. Following CapDec's training protocol, we trained on SentiCap and Flickrstyle10k datasets until the validation set loss reached a plateau. Additionally, we compared our results to the ZeroCap model (Tewel et al., 2021), which incorporates a style injection manipulation. We implemented three different manipulation techniques: (1) **ZeroCap + PM**, in which the style is injected into the LLM via prompting. We used the following prompts: for a positive style – "The beautiful image of a"; for a negative style – "The disturbing image of a"; for a humorous style – "The humorous image of a"; and for a romantic style – "The romantic image of a". (2) **ZeroCap + IM**, in which the style is injected via images into the CLIP model. We perform arithmetic operations on the input image embedding by adding the CLIP embedding of an emoji that represents the desired style (e.g., a smiley emoji for positive sentiment), and subtracting a neutral emoji embedding to discard the attributes belonging to the emoji itself. Finally, we implemented (3) **ZeroCap + IPM**, a combination of both aforementioned manipulation techniques. Although these methods are based on a zero-shot model, they require careful selection of prompts and images to achieve the desired style.

## 2.4 RESULTS

**Quantitative analysis** Table 1 shows our results for SentiCap (top table) and Flickrstyle10k (bottom table). The APOLLO-CAP-based models outperformed all baselines in terms of style accuracy across

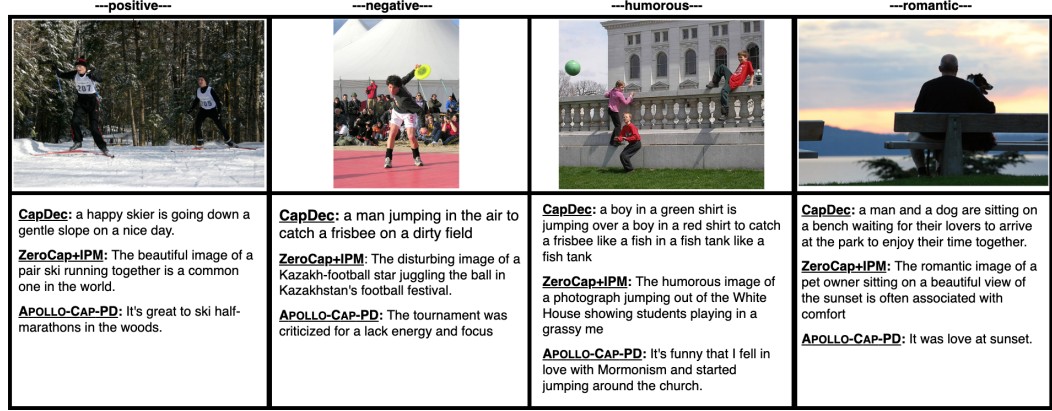

Figure 2: Examples of our APOLLO-CAP-PD compared to SOTA models.

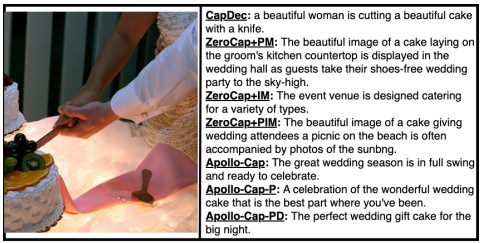

| | | last | beautiful | home | love |
|---|---|---|---|---|---|
| | GPT-2[(0)] | **0.036** | 0.0002 | 0.000 | 0.00005 |
| | CLIP[(0)] | 0.001 | 0.002 | 0.022 | **0.035** |
| | roBERTa | 0.000 | **0.019** | 0.000 | 0.002 |
| | CLIP* | 0.000 | 0.019 | 0.001 | **0.045** |
| | GPT-2* | 0.003 | **0.031** | 0.001 | 0.002 |

Figure 3: A positive-style caption example.      Figure 4: Models' token probabilities

all benchmarks. Although the ZeroCap-based approaches gained the highest TIC scores, they were partially successful in generating the required style, and in the qualitative test hereafter they performed the worst compared to the other approaches. The results also show that APOLLO-CAP-PD surpassed the state-of-the-art model, CapDec, in style accuracy on all styles, and in TIC on all styles except for the romantic style, while only slightly reducing the fluency score. It is important to note that this minor impact on fluency is acceptable, as a fluency score of 0.8 already indicates a good fluency level. Upon observing the results based on APOLLO-CAP, we can see that APOLLO-CAP-P and APOLLO-CAP-PD achieve significantly higher results in TIC and style accuracy, than APOLLO-CAP. APOLLO-CAP-PD outperforms APOLLO-CAP-P on TIC across all styles, but it is unclear which method APOLLO-CAP-P or APOLLO-CAP-PD performs better on the style accuracy. Additionally, the fluency scores for all of these approaches are sufficient, exceeding 0.8. The ZS methods based on APOLLO-CAP and ZeroCap exhibit larger vocabularies than the CapDec, which was trained on the task-specific dataset.

**Qualitative Analysis** In Figure 2 we present a comprehensive comparison of several approaches: APOLLO-CAP-PD (our leading approach), CapDec, and ZeroCap+IPM. We show results for the styles: positive, negative, humorous, and romantic. When comparing APOLLO-CAP-PD to CapDec, we observed that the former exhibits broader world knowledge in its captions, while the latter focuses mainly on technical details. For example, in the negative caption, APOLLO-CAP-PD identified the scene as a tournament, whereas CapDec provided drier factual information ("a man jumping..."). Moreover, CapDec used mainly common adjectives (e.g., 'dirty') to embed the style. In contrast, APOLLO-CAP-PD provided creative descriptions, such as 'criticized for a lack of energy', which contextualize the style within the narrative. This difference may be explained by CapDec mimicking the limited style displayed in training data, while APOLLO-CAP-PD leverages a general LLM that naturally implements styles in a storyline.

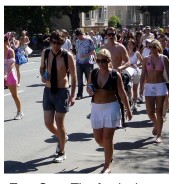 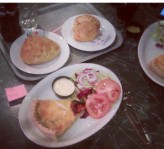 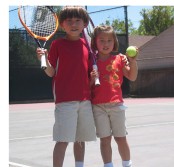 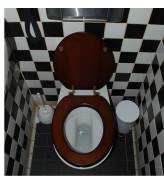 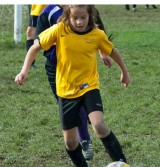 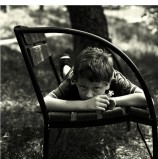

**ZeroCap:** The festival running from 18th to 21st of the first month is a marathon.
**APOLLO-CAP-P:** It's been a while since we've laughed about underwear parade stereotypes.

**ZeroCap:** This meal was delicious), and friends (L.
**APOLLO-CAP-P:** It is Breastfeeding Time.

**ZeroCap:** Itf A Mini 2008 I ________________Itf A Minil.
**APOLLO-CAP-P:** It's been a while since I've been laughed at by twins.

**ZeroCap:** The Chairroom WC 6.
**APOLLO-CAP-P:** This is a very simple baby-friendly and giggly place to laugh with your friends.

**ZeroCap:** In the final game of Manchester City's summer festival in January.
**APOLLO-CAP-P:** A kid giggling in the stands.

**ZeroCap:** A-Camp Kid - Flickr ÔòÀ AUSTIN COUNTY - Texas A-Campers are tired.
**APOLLO-CAP-P:** This son laughing at his father's solitude is a reflection of the way our society treats the lonely.

Figure 5: APOLLO-CAP-P caption examples for images and audio clips featuring children's laughter.

**Ablation Study** Figure 3 provides a comprehensive comparison of all approaches, illustrating a positive image caption. CapDec properly described the fact that a woman is cutting a cake and also added positive adjectives, yet it lacked real-world knowledge. In this case, it missed the celebration context. While ZeroCap's approach captured some relevant details they also exhibited instances of hallucination, as seen in examples like 'shoes-free wedding' and 'picnic beach'. APOLLO-CAP expressed the celebration of the wedding, and yet it dropped an important part of the content - the cake. In comparison, APOLLO-CAP-P included the important details - the celebration, the wedding, the cake, and relevant style adjectives; however, the fluency is degraded. Finally, APOLLO-CAP-PD met all the criteria - relevance ('wedding', 'cake' and 'night'), style ('perfect'), and fluency that is reflected in the proper integration of the content words.

**Model Ensemble Analysis** We provide a detailed explanation of the optimization process of APOLLO-CAP-PD for an input image of a vase with roses and a desired positive style. In this section, the notations $GPT^{(0)}$ and $GPT^{(*)}$, $CLIP^{(0)}$ and $CLIP^{(*)}$ refer to algorithms 1 and 2, respectively. The superscript zero denotes the expert model with its original context vector and the asterisk denotes the expert model with its optimized context vector. After generating the first token, 'The', Figure 4, displays the probability of the top-1 candidate tokens according to each expert. $GPT-2^{(0)}$ assigns the highest probability to 'last', while $CLIP^{(0)}$ identified 'love' as the next most likely token, possibly due to the common association of vases with roses in a romantic context. In contrast, roBERTa preferred 'beautiful' as the next token, possibly indicating its frequent use as a positive adjective to start a sentence. After one optimization step, $CLIP^{(*)}$ maintained its original preferences but also increased the probability of 'beautiful', aligning with both the desired style and the image. Finally, with the combined loss propagation of GPT-2, CLIP and roBERTa expert models, $GPT-2^{(*)}$ selected 'beautiful' as the token, which appears to be the most probable choice in this context. This suggests that APOLLO-CAP-PD considers various aspects, including style, image relevance, and text fluency.

## 3 AUDIO-AWARE IMAGE CAPTIONING

**Goal** We demonstrate our approach's ability to generalize to other modalities by introducing a novel task – Audio-Aware Image Captioning, that integrates audio into image captions. The input of the task is both an image and an audio clip, and the task is to generate text that describes the image within the context of the provided audio.

**Data** The test set contains 50 randomly sampled images from the Senticap (Mathews et al., 2016) test set, and the validation set contains five images from the Senticap validation set. For all images, we included an audio clip of children's laughter that we collected from (`https://freesound.org`).[1]

**Model** We adapted the stylized image captioning system of APOLLO-CAP-P (Section 2.2) by replacing the style component with an audio counterpart – CLAP (Wu* et al., 2023), which assess the correspondence between text and audio. We projected the audio clip and the candidate captions

---

[1]The data containing the images and audio is available at link-github

| Model | TAC | TIC | Fluency |
|---|---|---|---|
| APOLLO-CAP-P | 0.53 | 0.28 | 0.91 |
| ZeroCap | 0.08 | 0.32 | 0.81 |

Table 2: Averaged Scores for ZeroCap and APOLLO-CAP-P with laughter audio content on 50 images from the Senticap test set. TIC - text-image correspondence, TAC - text-audio correspondence

onto the CLAP embedding space and calculated the cosine similarity between each candidate and the audio embedding vectors. We then replaced the style probability in equation 6 with the audio probability and applied the rest of the algorithm without further changes.

**Qualitative Analysis**   Figure 5 shows examples of captions generated by APOLLO-CAP-P for images from the test set in the presence of audio featuring kids' laughter. APOLLO-CAP-P managed in all images to add the context of the audio – laughing. ZeroCap, which does not process audio, does not reflect this context. In the first left image APOLLO-CAP-P reasons that an image with many people walking is a parade and because they are with very little clothing he makes it into something funny (connected with the audio) – an 'underwear parade'. The second image humorously connects the image of a mother eating with the sound of a baby laughing, suggesting that the baby is eating as well, since the mother is eating – "It is Breastfeeding Time". Even when the image seems gloomy as in the rightmost image, the model manages to generate a caption that connects laughter with the negative sentiment of the image. These results demonstrate our method's ability to process audio and still show scene-level understanding.

**Quantitative Analysis**   Table 2 shows our APOLLO-CAP-P approach manages to get the text-audio correspondence (TAC), while maintaining the high fluency and text-image correspondence (TIC).

## 4   RELATED WORK

In recent years, there has been a shift in modeling towards transformer-based methods (Vaswani et al., 2017) which learn context and process sequential data through their attention mechanism. The next revolutions in machine learning came with the rise of *foundation models*, which are transformer-based models that have been injected with prior knowledge through pre-training on large datasets (Devlin et al., 2018; Lan et al., 2019; Yang et al., 2019; Zan et al., 2022; Kim et al., 2021; Zaheer et al., 2020; Baevski et al., 2020). These models have been shown to perform on various tasks, domains, and even multimodality (Liu et al., 2023; Yang et al., 2022) Their distinct capabilities mainly depend on their training data, for example, models that are trained on pairs of images and texts demonstrate capabilities in vision and language tasks (Tan & Bansal, 2019; Lu et al., 2019; Radford et al., 2021).

*Foundation* models' capabilities to perform in zero-shot have been utilized in *Socratic* approach which combines foundation models with frozen LLMs and bridges the gap through language, via prompting (Zeng et al., 2022; Tiong et al., 2022; Wang et al., 2022; Tsimpoukelli et al., 2021; Huang et al., 2023; Xie et al., 2022). In contrast to Socratic models, a different approach, that does not rely on prompting, guides the LLMs by tuning their prior knowledge in their attention mechanism with visual cues (Tewel et al., 2021). In our work, we present a generic approach for guiding multiple transformer models through gradient updates, which can be employed across different modalities.

## 5   CONCLUSIONS

We propose a modular framework that leverages the expertise of large pre-trained models and jointly solves complex tasks in a zero-shot setting without relying on prompting. Our approach enables decentralized control, allowing models to exchange expertise. We demonstrated our approach on two tasks. Our method achieves state-of-the-art results on two benchmarks for stylized image captioning. To demonstrate the method's capabilities, we tested its ability to work on audio, by introducing the novel task of audio-aware image captioning, in which an image and audio are given and the task is to generate text that describes the image within the context of the provided audio.

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

| Approach | Param | Pos | Neg |
|---|---|---|---|
| **APOLLO-CAP-PD** | $\tau$ | 0.14 | 0.17 |
| | $J$ | 1 | 1 |
| | $\lambda_{LM}$ | 0.22 | 0.61 |
| | $\lambda_{CL}$ | 1 | 2 |
| **APOLLO-CAP-P** | $\tau$ | 0.01 | 0.09 |
| | $\lambda_{LM}$ | 4 | 0.62 |
| | $\lambda_{CL}$ | 8 | 2 |
| **APOLLO-CAP** | $\tau$ | 0.01 | 0.01 |
| | $\lambda_{\text{CL}}$ | 2.2 | 8.1 |
| | $\lambda_{\text{SL}}$ | 9.7 | 10.5 |
| | $\lambda_{\text{LM}}$ | 2 | 2.9 |

Table 3: APPOLO-CAP HYPER-PARAMETERS USED FOR SENTICAP

## A APPENDIX

### A.1 DATASETS

We split Senticap into train, validation, and test subsets with a ratio of 0.57, 0.13, and 0.3 respectively. We ended up with a train set of 1,217 images, validation set of 265 images and test set listing 743 images.

In Addition, for the Flickrstyle10k dataset, similar to the approach in Nukrai et al. (2022) we used a split ratio of 0.75, 0.08, and 0.17, resulting in 4409 images for the training set, with 490 and 1000 images allocated for the validation and test sets, respectively.

### A.2 HYPER-PARAMETERS

For the ZeroCap approach we used the original hyper-parameters suggested by Tewel et al. (2021). The image arithmetic with the emojis worked best for us when multiplying the emoji embedding by 0.5. During all experiment with ZeroCap we applied 5 optimization steps to GPT-2, and searched over 5 beams as suggested by Tewel et al. (2021). For our APOLLO-CAP-based models, we adopted the hyper-parameter values from ZeroCap except of the loss weights, $\lambda$, which were tuned on 20 randomly selected images from the validation set of the target benchmark, SentiCap or Flickrstyle10k. In addition, the gradient descent step size $\alpha$ as well as the style softmax temperature were tuned on the same 20 selected images used for tuning the loss weight. We tuned hyper-parameters separately for each style to maximize the harmonic average of the TIC, style, and fluency metrics detailed in Section 2.3. For demonstrating decentralization, we implemented Algorithm 2 with gradient step size $\alpha = 0.3$. Table 3 details the hyper-parameters used in our experiments.

### A.3 DEEPMOJI

DeepMoji (Felbo et al., 2017) is pre-trained model which trained on millions of paired tweets-emojis in order to give an emotion assesment to a text. It generates a vector probability $\mathbf{r} \in R^{64}$. each index i represent the probability of emoji i to represents the text. The relevant emojis that this model is predict:

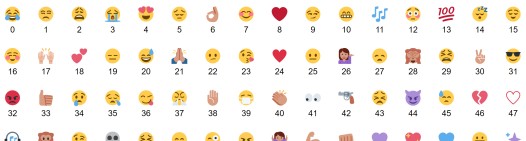 To represent a humorous style, we summed the probabilities associated with emoji indices {0, 53}. Conversely, for a romantic style, we focused on emojis with indices {4, 8, 18, 23, 24}.

