# OpenReview forum: "Apollo: Zero-shot MultiModal Reasoning with Multiple Experts"
_ICLR.cc/2024/Conference — Submitted to ICLR 2024_

### Official Review · Reviewer_6i3a · 2023-10-25

**Soundness:** 3 good
**Presentation:** 2 fair
**Contribution:** 2 fair
**Rating:** 3
**Confidence:** 4

**Summary:**

Apollo proposes a modular framework that integrates various expert modules and makes effort to improve the integration of these modules. Apollo does not require costly training expenses and can converge with just a few examples. The Authors conducted experiments on two downstream tasks: stylized image caption and audio-aware image caption.

**Strengths:**

1. Apollo's attempt to integrate several large foundational models into a single improved model is a significant endeavor, considering the background of various large language models that have emerged.
2. Apollo doesn't require an excessive amount of costly manual annotations.

**Weaknesses:**

1. Overclaiming of the contribution. First, the approach resembles knowledge distillation. It would be preferable to highlight the distinctions. Second, the paper's title claims that the model can achieve ZERO-SHOT MULTIMODAL REASONING, yet the experimental section only presents results for image captioning. This is insufficient since multimodal reasoning encompasses more meaningful tasks such as VQA, image-text retrieval, etc. Additionally, the paper claims that APOLLO can have 2 or more experts, but the experiments only include results for 3 experts. The paper should report results using just 2 experts, such as conventional zero-shot image captioning, and then progressively increase the number of experts. I'm also curious about how APOLLO handles 4 or more experts to accomplish multimodal reasoning tasks and the differences in using different numbers of experts for the same task. In summary, both the method's analysis and the experimental section lack in-depth insights and persuasiveness.

2. Unconvincing Experiments. Related to the first weakness, the paper only presents experimental results on two zero-shot image captioning settings, which lack persuasiveness. In the second setting, audio-aware image captioning, experiments were conducted on only 50 samples, which is too small, and all samples had laughing audio added, making the experimental setting overly uniform. It should include more variations in audio and sample comparisons and analysis. The paper also lacks comparisons with state-of-the-art baseline models, such as the MAGIC[1] model, which is another method that uses GPT-2 and CLIP for zero-shot image captioning. If APOLLO also uses only two experts, what are the comparative results? And what’s MAGIC’s performance on the style and audio captioning settings?

3. Lack of Model Efficiency Analysis. Since APOLLO uses multiple experts and requires multiple steps of optimization for each token prediction, the model's inference speed and computational cost will increase significantly, affecting its efficiency. The paper lacks experiments and comparison of model inference time and FLOPs.

**Questions:**

See weakness.

---

> ### Comment · Reviewer_6i3a · 2023-11-23
> **Summary of The Final Review**
>
> Post-rebuttal I think some limitations still exist since the experiments are unconvincing. So I'm going to keep my original score.

---

### Official Review · Reviewer_JWXZ · 2023-10-26

**Soundness:** 2 fair
**Presentation:** 2 fair
**Contribution:** 3 good
**Rating:** 5
**Confidence:** 3

**Summary:**

In this paper, the authors present, Apollo, a zero-shot method to improve generation of a language model with multiple expert models (including ones that can incorporate multimodal information), an extension from ZeroCap which only allowed one multimodal expert. The idea is to iteratively update the context of the language model using losses computed from the expert models. This approach can be used to generate stylized image captions using a GPT-2 model with 2 expert models: CLIP for image-text proximity and a text style classifier for style guidance. The authors showed that their method is able to achieve state-of-the-art results on stylized image captioning while remaining zero-shot. The approach is further examined on image captioning with accompanying audio, the authors showed that the resulting zero-shot generated captions reflect information from both audio and image.

**Strengths:**

The authors introduced a simple but effective approach of using expert models to guide an LM to perform certain tasks in a zero-shot manner. More importantly, the authors showed that this approach can be used for multimodal tasks (by having multimodal models as experts) and can incorporate guidance from multiple experts simultaneously. I believe this approach could have potential: with a stronger LM and better experts, we may be able to zero-shot more challenging multimodal tasks.

**Weaknesses:**

Some experiment design choices are questionable. For example, for the experiment in section 3, the only sound included is "children laughter", which is not enough to convince us that this approach will work on any other image/audio combinations. There should be at least a few more different audio for the result to be more convincing. Also, using models as both the expert model and the evaluation metric can be a little bit controversial, so maybe the authors should include some other metrics (such as human rating of relevance between the caption and image/audio).

It is unclear which of "Expert summation", "Expert product" and "Decentralization" works the best. The author introduced all 3 methods, but failed to provide any summary or conclusion on which one should be used. In Section 2 ablation study, the author claimed that "Decentralization" seemed to be more fluent and meets all criterion, but from Table 1 it seems that "Product" seems to have more fluency overall; moreover, in section 3, only Apollo-Cap-P (i.e. "Product") is used, thus make us question why "Decentralization" is needed/introduced.

Even though I believe that this approach could have potential in many challenging multimodal tasks if we can scale up the LM as well as the experts, the potential of the approach as well as possible future directions is not discussed in the paper.

The presentation of the paper needs improvement. For example,

(1) The term "lambda" in Eq(1) and Eq(2,4) seems inconsistent and represents different weights.

(2) The fonts of most Tables and Figures are quite small. Moreover, the fonts of Tables 1,2 and table in Figure 4 are all different and inconsistent. Also, in some tables (Table1 and Fig 4) the top performances are bolded, while in Table 2 there is no bolding.

**Questions:**

1. How is "expert model" defined? Conventionally, we would define the generating LM as the "learner" and the guiding models as "experts" (i.e. GPT-2 is learner and CLIP/CLAP/style model are experts). But in section 1.1, it seems like all transformers are "experts" (i.e. GPT-2 is an expert too).

2. How computationally costly is this model? Approaches that attempts to control LM generation outputs through nested-loop gradient descent are generally quite computationally expensive.

---

### Official Review · Reviewer_mH2v · 2023-11-06

**Soundness:** 2 fair
**Presentation:** 2 fair
**Contribution:** 2 fair
**Rating:** 5
**Confidence:** 3

**Summary:**

Paper investigates combination of multiple foundation models to solve multi-modal tasks.

Authors claim that while previous methods use language as intermediate representation to exchange information between different models, they propose to use a common latent space.

Experiments are conducted on two tasks, stylized image captioning and audio-aware image captioning to show the effectiveness of the approach.

**Strengths:**

1. Important Problem
- The problem tackled is very important. And it is going to become even more important going forward.

2. Method
- I overall like the method.
- Method section is written well too.
- I have a question regarding the method, which I have put in the questions box.

**Weaknesses:**

1. Regarding related work
- The authors claim that previous method use language to interface multiple models and authors propose to use latent space.
- LLava [1] also uses latent space to combine clip encoder to LLava. Can authors discuss this?
- Can authors also compare to LLava?

2. Writing
- A critical problem with the writing is that in the introduction authors are discussing a much broader problem. And they also motivate with some other problems 'However, the large and increasing variety of multimodal tasks (e.g., vision and language navigation (Ku et al., 2020), and video question answering (Lei et al., 2018)), do not have foundation models.'.
- But the experiments are not conducted for vision language navigation.
- So, I think it would be more fair if authors say that they are writing a paper about stylized image captioning, rather than over stating the problem, because they do not have results to back it up.
- Or the authors should make it clear in that same paragraph that they do not do experiments on these tasks.

3. Experiments concerns
- In the second experiment, I have following concerns:
- The dataset is too small: 50 image audio pairs. Maybe the authors can have an error bar, sample 50 images multiple times atleast
- The captions in Figure 5 (as discussed in page 9) do not make much sense to me, except for first image.

[1] Visual Instruction Tuning.  Haotian Liu, Chunyuan Li, Qingyang Wu, Yong Jae Lee

**Questions:**

One question I have is regarding the product of experts:
- Could you please elaborate on the text after equation 2. In particular, 'This operation merges the experts’ preferences and directs the transformer toward a common region, while maintaining proximity to the initial TM suggestion boundaries, as guided by the first loss term.'
- Can you do an ablation to confirm the motivation. Maybe without this term?

---

### Meta-Review · Area_Chair_ng22 · 2023-12-02

**Metareview:**

This paper presents Apollo, a modular framework that integrates various expert modules to solve multi-modal tasks. The authors mainly conducted experiments on two downstream tasks: stylized image captioning and audio-aware image captioning.

It received scores of 355. All the authors have some shared concerns about the paper: (1) over-claiming of the contribution, and (2) unconvincing experiments. The experiments are weak, and some experiment design choices are questionable. The presentation of the paper also needs improvement. No rebuttal is provided. Therefore, the AC would like to recommend rejection of the paper.

**Justification For Why Not Higher Score:**

All the authors gave reject recommendations, and no rebuttal is provided.

**Justification For Why Not Lower Score:**

N/A

---

### Decision · Program_Chairs · 2024-01-16

Reject